# Understanding data and information needs for palliative cancer care to inform digital health intervention development in Nigeria, Uganda and Zimbabwe: protocol for a multicountry qualitative study

Kehinde Okunade,[1] Kennedy Bashan Nkhoma,[2] Omolola Salako,[3] David Akeju,[4] Bassey Ebenso [iD],[5] Eve Namisango,[6] Olaitan Soyannwo,[7] Elizabeth Namukwaya,[8] Adlight Dandadzi,[9] Elizabeth Nabirye,[8] Lovemore Mupaza,[10] Emmanuel Luyirika,[6] Henry Ddungu,[11] Z Mike Chirenje,[9] Michael I Bennett,[12] Richard Harding [iD],[2] Matthew J Allsop[12]

For numbered affiliations see end of article.

**Correspondence to**
Dr Matthew J Allsop;
m.j.allsop@leeds.ac.uk

## ABSTRACT

**Introduction** Palliative care is a clinically and cost-effective component of cancer services in sub-Saharan Africa (SSA). Despite the significant need for palliative cancer care in SSA, coverage remains inadequate. The exploration of digital health approaches could support increases in the quality and reach of palliative cancer care services in SSA. However, there is currently a lack of any theoretical underpinning or data to understand stakeholder drivers for digital health components in this context. This project addresses this gap through engaging with key stakeholders to determine data and information needs that could be supported through digital health interventions.

**Methods and analysis** This is a multicountry, cross-sectional, qualitative study conducted in Nigeria, Uganda and Zimbabwe. In-depth interviews will be conducted in patients with advanced cancer (n=20), caregivers (n=15), health professionals (n=20) and policy-makers (n=10) in each of the three participating countries. Data from a total of 195 interviews will transcribed verbatim and translated into English before being imported into NVivo software for deductive framework analysis. The analysis will seek to understand the acceptability and define mechanisms of patient-level data capture and usage via digital technologies.

**Ethics and dissemination** Ethics approvals have been obtained from the Institutional Review Boards of University of Leeds (Ref: MREC 18–032), Research Council of Zimbabwe (Ref: 03507), Medical Research Council of Zimbabwe (Ref: MRCZ/A/2421), Uganda Cancer Institute (Ref: 19–2018), Uganda National Council of Science and Technology (Ref: HS325ES) and College of Medicine University of Lagos (Ref: HREC/15/04/2015). The project seeks to determine optimal mechanisms for the design and development of subsequent digital health interventions to support development, access to, and delivery of palliative cancer care in SSA. Dissemination of these findings will occur through newsletters and press releases, conference presentations, peer-reviewed journals and social media.

**Trial registration number** ISRCTN15727711

## Strengths and limitations of this study

► This study is the first to identify preferences of stakeholders involved in palliative cancer care provision in sub-Saharan Africa to inform digital health approaches.
► All aspects of the proposed study have been coproduced with experts in palliative care delivery and research in Nigeria, Uganda and Zimbabwe.
► This multicountry study will generate a logic model to target digital health approaches for palliative care that could have relevance across the sub-Saharan Africa region.
► This study restricts its focus to palliative cancer care so its relevance to other palliative conditions may be limited.
► The findings may not be reproducible beyond the three participating countries.

## INTRODUCTION

Due to late-stage clinical presentation, limited funding and restricted access to curative therapies, about 80% of cancers on the continent are incurable at the time of detection and diagnosis.[1] In 2018, there were over 770 000 new cancer cases and 514 000 cancer-related deaths across Eastern, Middle, Southern and Western Africa.[2] These figures are projected to continue to rise (up to 1.28 million new cases and 970 000 deaths) by 2030, with subsequent international and regional

political declarations constituting a new global non-communicable disease agenda.[3] Increasing incidence is attributed to factors that include ageing, high residual burden of infectious agents (HIV/AIDS, human papillomavirus, hepatitis B virus) and lifestyle factors in sub-Saharan Africa (SSA).[4] Our analysis reveals that by 2060, an estimated 16 million people with cancer will die annually with serious health-related suffering, a 109% increase between 2016 and 2060, with the fastest rise occurring in low-income countries (400% increase).[5]

Palliative care (PC)—the prevention and relief of physical, emotional, social or spiritual suffering associated with any chronic or life-threatening illness, from the point of diagnosis—is a vital and fundamental component of the basic and essential services within universal health coverage (UHC).[6] It is also a realistic response to support equitable, accessible and cost-effective interventions for cancer care in SSA. Independent of cancer prevention and treatment efforts in the region, PC remains a critical and essential component of care, with proven effectiveness and cost-effectiveness.[7 8] There have been enormous strides made in the development of PC services in the SSA region,[9 10] but there remains a need for significant expansion of provision to meet demand. Current provision of PC services is limited to 24 of 48 countries, up from only five in 2004, with only less than 5% of people who need PC being able to access services in the region.[10]

A major challenge to developing palliative cancer care across the African region is the lack of local evidence to ensure practice is evidence-based and replicable and reflects the needs of the population served.[6] Evidence to date has revealed that patients with advanced cancer in SSA have a high burden of physical and psychological symptoms, would prefer to have full information and better communication around their needs and care options, experience spiritual distress, their family caregivers face compounded poverty and psychological distress.[11–18] It is essential to create channels for gathering patient-level data as an indicator of quality as well as to inform clinical practice and audit. Furthermore, understanding how emerging services are supporting patients with advanced cancer through assessing experiences and outcomes is a priority for PC development in the region.[19] This can be achieved through the use of validated, context-specific tools for measuring outcomes for PC patients and their caregivers in SSA.[19 20] We have therefore developed a valid patient-reported outcome measure (PROM) for advanced disease in SSA to capture the core concerns of patients and families.[21–23] Capturing these data can enable development and adaptation of services to ensure they can respond to the specific needs of patients with cancer.

This protocol describes a study to explore how technology-based approaches could capture patient-level data from patients with advanced cancer that has utility across the health system. Previous work by our team has highlighted the potential of digital health to facilitate the collection, sharing and use of patient-level data. For example, we know that mobile phones are frequently used in multiple ways as part of PC service provision in the African region and that development of approaches that capitalise on mobile phones is a high priority for providers.[24] In SSA, mobile phone services are available to a larger portion of the population than many basic services (such as sanitation and financial services). Approaches using digital health can benefit from the widespread access and low cost of mobile phone devices in the region and have shown improved chronic disease management,[24] patient behavioural change and health systems strengthening,[25] reduced costs of patient monitoring, improved adherence and better communication. These benefits are greatest in rural areas.[26] Furthermore, such patient to provider telemedicine has recently been recommended by the WHO as an approach that can support health systems strengthening.[27]

In recent years, there has been exploratory research and development of digital health approaches in PC services in SSA.[28–30] However, this project seeks to address the lack of a theoretical underpinning to interventions using digital health components in this context. Our project will undertake engagement with key stakeholders (patient, caregivers, health professionals and policy-makers) across the health system to define the optimal mechanisms through which patient-level data, captured via digital health approaches, can be integrated into palliative cancer care delivery and improvement.

## RESEARCH OBJECTIVES AND QUESTIONS

The study aims to answer the question, what are the optimal mechanisms through which patient-level data, captured via digital health, can be used in the development and delivery of palliative cancer care in SSA?

The specific objectives of the project are to:

1. Establish a consortium of academic researchers (from the UK, Nigeria, Uganda and Zimbabwe), service user advocates, non-governmental organisations, PC providers, policy-makers and digital health development and implementation experts to catalyse digital health research and generate evidence that can guide palliative cancer care development across SSA.
2. Understand the acceptability and optimal implementation of patient-level data collection (eg, PROMs and patient-reported experience measures) using digital health approaches in Uganda, Nigeria and Zimbabwe through patient and caregiver engagement.
3. Determine information needs and pathways for leveraging evidence generated from digital health approaches in service development in Uganda, Nigeria and Zimbabwe through health professional and service manager engagement.
4. Determine information needs and pathways for leveraging evidence generated from digital health approaches in policy-making in Uganda, Nigeria and Zimbabwe through policy-maker engagement.

5. Define the mechanisms for the implementation of digital health approaches to support development of palliative cancer care in SSA.
6. Develop a theoretically informed logic model for implementing digital health approaches to improve PC in SSA.

## METHODS AND ANALYSIS

Through partnership with the African Palliative Care Association (APCA), the University of Leeds have assessed the use and priorities of digital health approaches in PC services in the African region.[28] Furthermore, King's College London, working with APCA, have been pioneering patient-level data collection in PC in SSA and developing PROMs[19 20]; simple checklists of symptoms and concerns that are widely adopted and enable staff, patients and families to identify main concerns and prevent suffering, maintain people at home, support families and optimise function. PROMs for people with serious incurable illness can improve care and patient well-being. This project will enable integration of these research initiatives, determining the architecture of digital technologies to facilitate uptake and utilisation of evidence-based approaches such as PROMs. This will take forward the science of digital health in this neglected field, enabling a logic model to be developed for subsequent evaluation and implementation.

### PC development in participating countries

The current development of PC for each participating country is summarised in table 1.

In recent years, policy reports of PC from participating study countries have commonly identified priority focus areas of improving access to pain medications, improving awareness by health professionals of the value and role of PC in supporting patients, and integration of PC with the existing public health system. A summary of policy documents and their key findings for each country is provided in online supplementary appendix A.

### Study design

This study will adopt a multicountry, cross-sectional, exploratory study using qualitative methods. In-depth interviews with patients, caregivers, health professionals and policy-makers will be used to understand the acceptability and define mechanisms of patient-level data capture and usage. The interpretation of the findings will define optimal mechanisms through which patient-level data, captured via digital health, can be used in the development, delivery and improvement of palliative cancer care in SSA. Alongside the planned research activities, a consortium focused on digital technology for palliative cancer care will be formed. This will include researchers from SSA and the UK, alongside key stakeholders in each of the participating countries (ie, relevant policy representatives from ministries of health, civil society, patient advocates and digital health specialists) to understand: (1) current digital health development in Uganda, Nigeria and Zimbabwe; (2) opportunities for capacity development around digital health in palliative cancer care and (3) routes to uptake and translation of findings from planned research activities.

### Study participants

The study participants will be adults living with advanced cancer, their caregivers, their health professionals and policy-makers with a focus on cancer, non-communicable diseases and/or technology. Patients will include adults with advanced cancer receiving PC. Caregivers will include those supporting palliative cancer patients receiving care from recruiting facilities. Health professionals will be drawn from the clinical teams associated with study partner institutes alongside related services delivered in the locality. Policy-makers will comprise district or national level policy-makers working within cancer, non-communicable diseases or digital health. Table 2 outlines the inclusion/exclusion criteria, sampling characteristics and sample size for the four stakeholder groups.

### Data collection and management
#### Patients

Clinical staff at recruiting facilities will be responsible for identifying participants. These staff will be asked to verify that patients are aware of their PC diagnosis. This will be through review of clinical records and discussion with the participant's health professionals by the research assistant. Patients who are deemed ethically inappropriate by members of the clinical team, for example, where death is imminent, will not be approached. Participating patients will only be asked to participate in one face-to-face interview. The location of the interview will be decided by the patient or caregiver (ie, meeting either at their home, following a clinic appointment at a health facility, on the ward or at a neutral location, dependent on the patient's preference and the patient's clinical management at the time of the interview). Independent of the location chosen, a quiet and private room will be recommended to the patient for the interview. Arrangements will be made for such space at clinics and hospices ahead of interviews. A topic guide will be used to direct semistructured interviews with patients. The topic guide will explore current interaction with and access to PC services, their use of technology, the acceptability of using digital technology approaches to support interaction with health services and data collection, the clinical response anticipated from health services (with and without facilitation by digital health interventions) including perceptions of effective responses and alternative approaches to patient-level data collection without mobile phones.

#### Caregivers

Caregivers of patients with advanced cancer will be identified and approached to participate by clinical staff at recruiting facilities. Caregivers will only be required to participate in one face-to-face interview. A topic guide will be used to direct semistructured interviews with

| Table 1 | Summary of palliative care (PC) development in Nigeria, Uganda and Zimbabwe |
|---|---|
| **Country** | **Summary of development** |
| Nigeria | In Nigeria—the most populous country in SSA—PC is disparately spread in centres across the country. Historically, PC development in western Africa has been secondary to developments in the eastern and southern parts of the continent, and this continues to be the case. This is due mainly to the relatively low HIV prevalence, which meant that the region did not qualify to receive funding from the US President's Emergency Plan for AIDS Relief (PEPFAR) in the early 2000s.[32] Widespread interest in PC development in Nigeria emanated from the pain and suffering witnessed by concerned health professionals in patients with advanced cancer. Early efforts by the Hospice Nigeria team included an advocacy visit by Anne Merriman (the founder of Hospice Africa Uganda in 1993) which yielded little result as there was no opioid analgesics in the country to manage the associated cancer pain.[33] The current movement that has resulted in establishment of holistic PC services across Nigeria commenced in 1996 with the Ibadan 'Cancer pain group'.[34] Concerted advocacy and other activities of the group facilitated importation of opioid analgesics by the Federal Ministry of Health for the management of severe pain by 2001 and morphine powder for oral morphine preparation by 2005. |
| | The first PC team was founded in 2003 at Ibadan, providing both hospital-based and home-based services,[32] and the Hospice and PC Association of Nigeria was created in 2007. Members, most of whom received PC initiator's training at Hospice Africa, Uganda, function as individuals or teams across the six geopolitical areas of the country. Home-based PC continues as a very important form of PC delivery, with evidence of its benefits to patients and their families.[35] Some aspects of PC are now included in the national guidelines for HIV and AIDS treatment and care in adolescents and adults (2012) as well as the national cancer control plan of the Federal Ministry of Health (2018–2022). |
| | Current barriers to PC development include lack of government guidelines, poor knowledge about PC importance at all levels—policy-makers, public, health professionals, lack of inclusion of PC in curricula of health professionals and in the national health budget and National Health Insurance Scheme, poor availability and accessibility of strong opioid analgesics for cancer pain management. However, since 2012, the 'Treat Pain project' and 'Pain Free Hospital Initiative' of the Federal Ministry of Health, Global Access to Pain Relief Initiative and American Cancer Society have improved the opioid situation.[36 37] More can still be achieved through the training of more health professionals and change of attitude, increased funding and increased public awareness of the services.[35] There has been some early context setting work suggesting willingness to explore the application of technology within PC services, such as the use of telemedicine.[38] |
| Uganda | The Atlas of Palliative Care Development in Africa ranked Uganda highly in terms of palliative service development and its integration into the health system.[39] Prior to this, The Economist Intelligence Unit's Quality of Death Index[40] which is a 'measure of the quality of PC provided to adults in over 80 countries', ranked Uganda as 35th in the world. This index gives a general impression of the quality of life at the end of life and recognises continued development of PC services in Uganda. In terms for service delivery, Uganda has over 229 PC service outlets, for a population of 40 million people. These include hospices, home-based care and health facility-based outlets. The level of paediatric PC service development remains poor, with two paediatric PC service centres despite a large population of children and young people.[1] |
| | In terms of the wider health system, Uganda has been progressing the presence and development of PC services. Uganda is building a critical mass for PC professionals with PC integrated into the medical and nursing and is a hub for training and education in PC, with diploma, certificate and degree courses in PC provided by universities, specialised PC institutions like Mildmay Uganda, and international organisations like African Palliative Care Association (APCA). In terms of health information systems, there are two national indicators of PC into the national electronic health information system: patients presenting with pain and those receiving morphine for pain management. With this advancement, to some extent, performance of PC can be monitored at the national level. |
| | Access to pain medication is a crucial component of PC delivery in Uganda. The availability of the full analgesic ladder and palliative medicines remains poor,[41] compromising care providers' ability to support management of symptoms and alleviate suffering associated with pain. However, Uganda has introduced prescribing across cadres other than doctors, such as PC nurse prescribers. Furthermore, the rolling out of local production of morphine within Hospice Africa Uganda led to significant reductions in costs associated with morphine procurement.[42] This key opioid for moderate to severe pain is now provided free of charge and is supplied through the national supply chain mechanisms of the country. Although oral morphine is free, access still remains a challenge because of several factors such frequent stock outs, reluctance to prescribe morphine for fear of addictions as well as limited access to this medicine at primary healthcare level.[29] |
| | PC is included in the national health strategic plan, and in the HIV care guidelines. A stand-alone PC policy was drafted in 2017 and is pending cabinet approval. It is envisioned that a stand-alone policy will be a strong pillar in giving a strategic direction to service development and evaluation to provide a comprehensive framework for the integration of PC services into the healthcare system. It will also be used as tool to advance financing for PC advocacy. |

Continued

| Table 1 | Continued |
|---|---|
| **Country** | **Summary of development** |
| Zimbabwe | In 2014, a WHO report identified 1 in 60 Zimbabweans need PC,[43] alongside an International Children's Palliative Care Network report identifying a significant need among children across multiple provinces in Zimbabwe.[44] Despite the continuing high level of PC need, Zimbabwe was the first country in sub-Saharan Africa to have a hospice, founded in 1979. In terms of development of PC across the country, by 1997 there were 17 regional branches that had been formed throughout the country with 13 organisations providing PC by 2004.[45] The disease focus of PC in Zimbabwe was initially cancer, although the growth of the disease burden due to HIV and AIDS led to widening of provision to include those living with HIV and AIDS and other chronic illnesses. Notable initiatives have facilitated PC services provision in the country. In 1992, the Ministry of Health and Child Care formed the Prevention and Control of Cancer Committee in Zimbabwe that comprised relevant stakeholders and professionals. The committee oversaw the development of a 10-year National Cancer Control Programme Plan for Zimbabwe (1994–2004), with the overall aim to formulate, plan and implement a coordinated and cost-effective programme for the prevention and control of cancer in Zimbabwe. This led to the establishment of PC training across eight provinces and two cities of Harare and Bulawayo. The APCA in 2010 reviewed national policy documents and implementation guidelines from 10 Southern African countries, including Zimbabwe.[46] For Zimbabwe, while PC was highlighted as a priority area across documents, issues around minimal coverage of PC was noted alongside an absence of detail relating to opioid availability. In more recent years, the Worldwide Hospice Palliative Care Association global update of mapping levels of PC provision in 2014 placed Zimbabwe in category 4a.[47] Category 4a suggests a country as hospice-PC services at a stage of preliminary integration into mainstream service provision. This category suggests the development of a critical mass of PC activism in a number of locations, a variety of PC providers and types of services, awareness of PC on the part of health professionals and local communities, the availability of morphine and some other strong pain-relieving medicines, limited impact of palliative care on policy, the provision of a substantial number of training and education initiatives by a range of organisations, and interest in the concept of a national PC association. |

caregivers. The themes addressed in the topic guide will align with the patient topic guide.

### Healthcare professionals

Health professionals will be identified by clinical leads in each of the three countries. Through existing networks of PC providers in each of the countries, the clinical lead will approach health professionals to participate in the study. A topic guide will be used during health professional interviews. The topic guide aligns with stages of the data-use conceptual framework[15]: data demand (eg, current availability, use and quality of data for clinical decision-making), data collection (eg, feasibility of digital technology approaches to patient-level data collection), data availability (eg, clinical triggers in management of patients with advanced cancer, capacity to respond to information, information needs to inform patient care) and data utilisation (eg, sharing and accessing data via digital health approaches, data reporting priorities to regional and national health authorities). Causal elements linked to organisational, technical and behavioural factors influencing data use will be explored for each stage.

### Policy-makers

Policy-makers will be identified and approached by the APCA alongside academic and clinical teams in Nigeria, Uganda and Zimbabwe. A topic guide for key informant interviews will address access to and use of evidence to inform decision-making, seek comment on findings from a desktop review of existing policy on digital health to be conducted prior to key informant interviews, preferred mode and presentation of data, frequency of data reporting needed to inform decision-making on financing of PC services and on accelerating UHC.

Interviews across all recruitment sites in Uganda, Nigeria and Zimbabwe (outlined in online supplementary appendix B) will be undertaken by research assistants, supervised by the academic partners in Nigeria (KO), Uganda (ENamu) and Zimbabwe (MC). All interviews will be audio-recorded. Data security will be ensured through use of password-protected file sharing using the Microsoft OneDrive platform. Separate folders will be created for each country, with oversight from the lead institution, the University of Leeds. Only research team members will have access to the folder. All members of the project team will sign a data sharing agreement outlining explicit guidance regarding handling and management of research data that takes account of both the funders and national research council guidelines. On completion of this study, all electronic data on the OneDrive folder will be moved to an electronic archive for 5 years prior to being permanently destroyed. Deidentified research data deemed suitable for sharing will be hosted by Research Data Leeds, the institutional research data repository for the University of Leeds. All study participants will be assigned an identification code, which will be delinked from their identity at data entry point.

### Reflectivity

The research team comprises experts in digital health intervention development (MJA, BE), qualitative research (DA, AD, ENamu), global PC provision (KBN, ENami, EL, RH), monitoring and evaluation (LM), health services research in SSA (MC) alongside consultants in palliative medicine (ENamu, HD, OS, MIB) and oncologists (KO, OS). None of these researchers have any relationship with the patients or caregivers who will be approached to participate. However, a small number of the healthcare professionals or policy-maker participants may be known contacts.

### Patient and public involvement

The rationale for this work arose from priorities identified through surveying a key stakeholder group,

**Table 2** Population and sampling criteria for the four stakeholder groups

| Participant group | Eligibility criteria | Sampling criteria | Target number to recruit in each participating country |
|---|---|---|---|
| Patient | **Inclusion criteria:**<br>▶ Patients with advanced cancer (defined as those with metastatic cancer (where, if possible, determined through histological, cytological or radiological evidence) and/or those receiving anticancer therapy with palliative intent receiving palliative care<br>▶ Aged ≥18 years<br>**Exclusion criteria:**<br>▶ Patients with significant cognitive impairment that prevents informed consent<br>▶ Inadequate physical and mental health of a patient prior to recruiting (as deemed by the identifying clinician)<br>▶ Lack of a shared language between researcher and respondent<br>▶ Below 18 years of age | ▶ Age<br>▶ Sex<br>▶ Cancer type<br>▶ Location at the time of interview (community based or ward based) | ▶ 20 purposely selected patients |
| Caregiver | **Inclusion criteria:**<br>▶ A primary caregiver/family caregiver of a patient with advanced cancer who is at least 18 years of age<br>▶ Primary caregiver confirmed by the patient, including those who provide unpaid, informal provision of one or more physical, social, practical and emotional tasks. In terms of their relationship to the patient, they may be a friend, partner, ex-partner, sibling, parent, child or other blood or non-blood relative<br>▶ Both caregivers of participating and non-participating patients will be recruited into the study<br>**Exclusion criteria:**<br>▶ Below 18 years of age<br>▶ Inadequate physical and mental health of a patient prior to recruiting (as deemed by the identifying clinician) | ▶ Sex<br>▶ Age<br>▶ Patient involvement in study (ensuring representation of those where the patient whom they care for has participated, and those where the patient has not participated) | ▶ 15 purposely selected caregivers |
| Health professional | **Inclusion criteria:**<br>▶ Working with palliative care patients<br>▶ At least 6 months working experience at recognised palliative care facilities | ▶ Role (doctor, clinical officer, nurse, social worker, psychologist, pharmacist)<br>▶ Typical work setting (ie, community based, ward based) | ▶ 20 purposely selected health professionals |
| Policy-maker | **Inclusion criteria:**<br>▶ Representative from government ministry or national association responsible for oversight and development of healthcare in participating country | ▶ Working at different levels of the health system (district, national)<br>▶ Policy remit of their post (cancer, non-communicable diseases, digital health) | ▶ Up to 10 purposely selected policy-makers |

healthcare providers delivering PC across SSA.[24] Patients and caregivers were not involved in the design of the study. However, alongside research activities, the team will develop a consortium which will include patient advocates. This forum will be used to explore routes for communicating study findings to patient groups and help to establish potential routes for identifying patient and caregivers, or advocates, in subsequent projects.

### Study dates

November 2018–December 2019. Policy-maker engagement will begin in November 2018, with data collection commencing in March 2019.

### Data analysis

Interviews will be transcribed verbatim and translated into English before being imported into NVivo software

for deductive framework analysis.[31] An initial charting of pseudonymised transcripts (KBN) will be developed through line-by-line coding. This initial code will then be applied to a random set of transcripts (n=3) in each stakeholder group by three further members of the research team (Nigeria, Uganda, Zimbabwe), and the final framework agreed through discussion with the wider team. Once inconsistencies are resolved, the joint coding frame will be used to code all interviews across the participant group. Comparative analysis in the framework will enable us to identify common themes as well as country-specific and stakeholder group divergences. A model of the coding frame will be developed, and each theme and subtheme given a definition to ensure internal consistency of each code. Illustrative codes will be reported for each theme, with the study ID code to demonstrate reporting from across the sample breadth. In the final project meeting we will hold a Theory of Change workshop to model the planned digital health pathway within its context, detailing processes, stakeholder roles, outcomes and intended impact.

## Ethical considerations

Ethics approvals have been obtained from the Institutional Review Boards of University of Leeds (Ref: MREC 18–032), Research Council of Zimbabwe (Ref: 03507), Medical Research Council of Zimbabwe (Ref: MRCZ/A/2421), Uganda Cancer Institute (Ref: 19–2018), Uganda National Council of Science and Technology (Ref: HS325ES) and College of Medicine University of Lagos (Ref: HREC/15/04/2015). Ethical review undertaken by all project investigators ensured standard processes (dignity, autonomy, informed consent, confidentiality, anonymity, ability to adhere to protocol) and data security were considered in the protocol development. The project will be aligned with the Medical Research Council good research practice guidelines and H3Africa framework for conducting ethically responsible biomedical research. With reference to research on PC populations, a patient's condition should not preclude them from participation. Instead, additional strategies of harm minimisation have been developed collaboratively by the research team. Academic leads and coinvestigators will support protocol adaptation for local cultural appropriateness (eg, modifying recruitment accounting for cancer-related stigma). Researchers will routinely contact clinical teams to check the health status of a patient prior to contact for research activities. In terms of data governance, project documentation and deidentified data for joint analysis will be shared via a secure IT infrastructure hosted by the University of Leeds. Voluntary and informed participation, confidentiality and safety of participants will constitute key principles of researcher–respondent interaction. Written consent or a thumb print will be obtained from patients, caregivers, healthcare professionals and policy-makers prior to their enrolment in the study.

## Dissemination

We will provide an understanding of the mechanisms by which digital health approaches can facilitate evidence generation and use, ensuring optimal implementation with clear pathways for integrating captured data into existing health system functions. On completion of planned research activities, we will have:

1. Instigated an active, interdisciplinary consortium focused on technology-based approaches to developing palliative cancer care in SSA.
2. Defined mechanisms for optimal implementation of digital health interventions to support PC service development in SSA.
3. Provided a list of factors to target with digital health approaches with accompanying programme theory.
4. Obtained essential preliminary data needed to inform future research in digital health technology development for PC in SSA.
5. Developed a logic model for implementation of digital health to improve advanced cancer care in SSA.
6. Strengthened capacity for research, innovation and knowledge exchange in partner institutions and identified future capacity building needs.

Undertaking primary research with patients, caregivers, health professionals and policy-makers is essential to guide digital health approaches for palliative cancer care in SSA. A number of activities will be undertaking to support dissemination of project findings. These will include: (1) developing newsletters and press releases to communicate key project findings to the general public, (2) developing a dedicated website for the study where results will be publicly accessible by national and international policy-makers, practitioners and academics, (3) delivering presentations at local and national conferences in participating countries, alongside presentation at international conferences, (4) publishing articles in peer-reviewed journals and (5) social media through research team member and institutional accounts. Participants will be anonymised in any dissemination activities. Only pseudonymised, non-identifiable characteristics and quotes will be used in dissemination.

## CONCLUSION

This paper reports the protocol for a cross-sectional study with qualitative methods aimed at understanding the optimal mechanisms through which patient-level data, captured via digital health, can be used in the development, access and delivery of quality palliative cancer care in Uganda, Nigeria and Zimbabwe. The findings of the study will be device agnostic, providing a theoretical framework that can be used to inform a wide range of digital health intervention development and implementation. Without this research there is a risk of digital health intervention development for PC occurring in silos across SSA that do not take account of the multiple uses and value of data for stakeholders across the wider health system. Mapping information and data needs

across PC services will also create multiple opportunities for research. This includes subsequent piloting and evaluation of digital health interventions and validation of the data they capture, development of patient-focused digital health interventions such as information provision and self-management support, and exploration of the influence of factors such as gender, intersectionality, disability and cancer type on utilisation and engagement with digital health approaches. Subsequent development of digital health approaches for PC in SSA, gathering patient-level data and facilitating patient-provider communication, could lead to multiple benefits for patients and caregivers (reduced costs associated with time and travel to facilities, extend coverage and reach of services such as rural areas with mobile connectivity), health professionals (ability to identify and respond to specific and rising demand from patients with cancer) and policy-makers (receiving appropriate and timely data to inform service planning, guide integration of PC with wider healthcare delivery and contribute to strengthening of national digital health systems). There is an added imperative to understand how best to use digital health technologies for those receiving and providing care for advanced disease in SSA. There is currently a lack of evidence on the preferences of patients with advanced disease, their caregivers and their health professionals. Understanding these preferences is crucial to inform digital health intervention development, such as modelling interventions around the preferences of the timing and delivery of digital health approaches in PC.

**Author affiliations**
[1]College of Medicine, University of Lagos, Lagos, Nigeria
[2]Florence Nightingale Faculty of Nursing Midwifery and Palliative Care, Cicely Saunders Institute, King's College London, London, UK
[3]Department of Radiation Oncology, Lagos University Teaching Hospital, Lagos, Nigeria
[4]Department of Sociology, University of Lagos, Lagos, Nigeria
[5]Nuffield Centre for International Health and Development, Leeds Institute of Health Sciences, University of Leeds, Leeds, UK
[6]African Palliative Care Association, Kampala, Uganda
[7]Centre for Palliative Care, University College Hospital, Ibadan, Nigeria
[8]Department of Internal Medicine, Makerere University, Kampala, Uganda
[9]Clinical Trials Research Centre, College of Health Sciences, University of Zimbabwe, Harare, Zimbabwe
[10]Island Hospice and Healthcare, Harare, Zimbabwe
[11]Uganda Cancer Institute, Kampala, Uganda
[12]Academic Unit of Palliative Care, Leeds Institute of Health Sciences, University of Leeds, Leeds, UK

**Acknowledgements** We thank Olasupo Oyedepo, Director of the African Alliance of Digital Health Networks and Project Director at ICT4HEALTH Project for guidance on the development and implementation of this project.

**Contributors** MJA conceived the study; MJA, BE, RH, ENami, ENamu and MC contributed to the development of the study design and final protocols for sample selection, observations and interviews; KO, MJA, OS and ENami developed a draft of the manuscript; All authors contributed to writing the manuscript.

**Funding** This work was supported by the Medical Research Council (grant no MR/S014535/1) and Research England Quality-Related Global Challenges Research Fund (QR GCRF) through the University of Leeds.

**Competing interests** None declared.

**Patient consent for publication** Not required.

**Ethics approval** Ethics approvals have been obtained from the Institutional Review Boards of University of Leeds (Ref: MREC 18-032), Research Council of Zimbabwe (Ref: 03507), Medical Research Council of Zimbabwe (Ref: MRCZ/A/2421), Uganda Cancer Institute (Ref: 19-2018), Uganda National Council of Science and Technology (Ref: HS325ES) and College of Medicine University of Lagos (Ref: HREC/15/04/2015).

**Provenance and peer review** Not commissioned; externally peer reviewed.

**Data availability statement** No additional data available.

**ORCID iDs**
Bassey Ebenso http://orcid.org/0000-0003-4147-0968
Richard Harding http://orcid.org/0000-0001-9653-8689

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
