## [Reviewer comments · BMJ Open]

ARTICLE DETAILS

TITLE (PROVISIONAL)	Understanding data and information needs for palliative cancer care to inform digital health intervention development in Nigeria, Uganda and Zimbabwe: protocol for a multi-country qualitative study
AUTHORS	Okunade, Kehinde; Bashan Nkhoma, Kennedy; Salako, Omolola; Akeju, David; Ebenso, Basse; Namisango, Eve; Soyannwo, Olaitan; Namukwaya, Elizabeth; Dandadzi, Adlight; Nabirye, Elizabeth; Mupaza, Lovemore; Luyirika, Emmanuel; Ddungu, Henry; Chirenje, Z. Mike; Bennett, Michael; Harding, Richard; Allsop, Matthew

VERSION 1 - REVIEW

REVIEWER	Dr Bhawna Sirohi Dr Bhawna Sirohi , Consultant Medical Oncologist – GI & Breast Cancers Director, Medical Oncology – Max Institute of Cancer Care, New Delhi Secretary- EBMT Committee on Nuclear accidents President, Oncology Section – Royal Society of Medicine Honorary Consultant, Queen Mary University London, UK Mobile : +91 9756999976 & +44 7468520613 Email: Bhawna.sirohi13@gmail.com Bhawna.sirohi@maxhealthcare.com Twitter @SirohiBhawna
REVIEW RETURNED	07-Aug-2019

GENERAL COMMENTS	this is a well written study to assess the needs of palliative care in sub-Saharan Africa. this study will hopefully inform the key stake holders about future provision of services. this study is likely replicable in Low and middle income group countries.
---

REVIEWER	Megan E Doherty Children's Hospital of Eastern Ontario, Canada
REVIEW RETURNED	31-Aug-2019

GENERAL COMMENTS	Clear and well designed protocol.
-----------------------------------

REVIEWER	Felix Mühlensiepen Medizinische Hochschule Brandenburg Theodor Fontane, Germany
REVIEW RETURNED	17-Sep-2019

GENERAL COMMENTS	The study protocol describes a multy-country qualitative study which aims to explore the use of digital health in palliative cancer care in SSA. The results of the described study could make a very important contribution to the quality and reach of palliative care in SSA and beyond. The care situation in the three countries in which the study is to be implemented is described very precisely and comprehensibly. The aims of the study are clearly defined in the manuscript. The study design is suitable for the exploration of the research object. As far as this can be assessed based on a manuscript, the authors fulfil the ethical requirements for such a research project. The authors pursue a coherent dissemination strategy. However, some aspects of the manuscript still need to be adapted before the article can be published: General So far, no study dates are mentioned neither in the abstract nor in the rest of the manuscript. Please indicate the study dates, preferably also with regard to the individual work packages. Abstract Please add the number of interview partners. Please describe the analysis of the interviews. The last sentence of the Methods and analysis section is not method-related and can be removed. Methods and analysis Table 1 describes the development of palliative care in the three countries very well and specifically. In my view, this table does not belong in the Methods and analysis section, but rather in the Introduction. Since this would then be very extensive, I suggest to add the table to the appendix. I also suggest two minor changes in the table: p. 9, l. 4: "free of change"; p. 10, l. 3: a dot is missing here. I hope the recommendations are useful to the authors. I believe that if the authors address these comments the study protocol will improve greatly and can be accepted for publication.
--

VERSION 1 – AUTHOR RESPONSE

Reviewer comments	Response from authors
Reviewer 1	
this is a well written study to assess the needs of palliative care in sub-Saharan Africa. this study will hopefully inform the key stake holders about future provision of services.	Many thanks for your positive and helpful feedback.

this study is likely replicable in Low and middle income group countries.	
Reviewer 2	
Clear and well-designed protocol.	Thank you for your succinct feedback on the protocol.
Reviewer 3	
The study protocol describes a multi-country qualitative study which aims to explore the use of digital health in palliative cancer care in SSA. The results of the described study could make a very important contribution to the quality and reach of palliative care in SSA and beyond. The care situation in the three countries in which the study is to be implemented is described very precisely and comprehensibly. The aims of the study are clearly defined in the manuscript. The study design is suitable for the exploration of the research object. As far as this can be assessed based on a manuscript, the authors fulfil the ethical requirements for such a research project. The authors pursue a coherent dissemination strategy.	Many thanks for your positive and helpful feedback. We outline how we have responded to your helpful feedback below.
General: So far, no study dates are mentioned neither in the abstract nor in the rest of the manuscript. Please indicate the study dates, preferably also with regard to the individual work packages.	We have added the overall study dates and start dates for policymaker engagement and data collection to the manuscript at the end of the data collection section.
Abstract: Please add the number of interview partners. Please describe the analysis of the interviews. The last sentence of the Methods and analysis section is not method-related and can be removed.	We have added the intended number of participants in each stakeholder group to the abstract, alongside adding details of the analysis approach. We have also removed the last sentence from the methods and analysis section of the abstract and incorporated this into the dissemination section.
Methods and analysis: Table 1 describes the development of palliative care in the three countries very well and specifically. In my view, this table does not belong in the Methods and analysis section, but rather in the Introduction. Since this would then be very extensive, I suggest to add the table to the appendix.	As you highlight, the table does include specific development of palliative care in each of the three countries of the study providing useful context specific to the sites where the research is taking place. We feel that this is important data for the protocol that we could want to remain in the main body of the manuscript. As we understand it, tables are

	typically not displayed in full in the online published version, but can be clicked on and expanded by the reader so will not detract from the surrounding text or make the section appear too extensive. We have discussed whether Table 1 could be added to the introduction section, although we cannot find where it would be suitably placed. If you are happy to allow it, we would request that Table 1 remains in the methods and analysis section or guidance is provided on its appropriate placement in the introduction section.
I also suggest two minor changes in the table: p. 9, l. 4: "free of change"; p. 10, l. 3: a dot is missing here.	Thank you for such meticulous review of the manuscript and for highlighting these errors. We have now addressed this in the manuscript.